# The *Bmtret*1 Gene Family and Its Potential Role in Response to BmNPV Stress in *Bombyx mori*

**DOI:** 10.3390/ijms25010402

**Published:** 2023-12-28

**Authors:** Mingjun Lin, Yixuan Qian, Enxi Chen, Mengjiao Wang, Gui Ouyang, Yao Xu, Guodong Zhao, Heying Qian

**Affiliations:** 1Jiangsu Key Laboratory of Sericultural Biology and Biotechnology, School of Biotechnology, Jiangsu University of Science and Technology, Zhenjiang 212100, China; lmj06062023@163.com (M.L.); qianyx0915@163.com (Y.Q.); 212211803101@stu.just.edu.cn (E.C.); wangmengjiao001128@163.com (M.W.); yaoxu208@163.com (Y.X.); 2Key Laboratory of Silkworm and Mulberry Genetic Improvement, Ministry of Agriculture and Rural Affairs, The Sericultural Research Institute, Chinese Academy of Agricultural Sciences, Zhenjiang 212100, China

**Keywords:** *Bombyx mori*, *Bmtret*1, BmNPV resistance, bioinformatics analysis, viral replication

## Abstract

Trehalose is a non-reducing disaccharide and participates in physiological activities such as organ formation, energy metabolism, and stress resistance in insects. The *Bmtret*1 gene family is mainly involved in in the sugar metabolism of silkworm. In the present study, phylogenetic analysis divided 21 *Bmtret*1 orthologs into three clades. These genes are equally distributed on the nine chromosomes. The cis-elements in the promoter regions of *Bmtret*1s indicated the possible function of *Bmtret*1s in response to hormones and environmental stimulus. The qPCR analysis showed the significantly different expression levels of *Bmtret*1s in different tissues and organs, indicating possible functional divergence. In addition, most *Bmtret*1s showed disturbed expression levels in response to silkworm nuclear polyhedrosis virus (BmNPV) stresses. Our results provide a clue for further functional dissection of the *Tret*1s in *Bombyx mori* and implicate them as potential regulators of antiviral responses.

## 1. Introduction

The silkworm, considered a model invertebrate creature, was the first insect used for silk production in human history and was widely used throughout domestication [1]. However, many viral diseases can pose a serious threat to the growth and development of silkworm [2,3]. Silkworm nuclear polyhedrosis virus (BmNPV) infection is a major threat to sericulture and can cause serious economic losses [4,5,6]. When BmNPV infects a host, two types of virus particles are produced: early budding virions (BVSs), which are transmitted primarily between cells, and late occlusion-derived virions (ODVs), which are transmitted primarily between hosts [7,8]. ODV virus particles are packaged in the polyhedra of a highly symmetric covalent crosslinked lattice [9]. BmNPV mainly infects silkworm larvae through the mouth. The polyhedron is alkaline-lysed by the host intestinal environment, and the enteric membrane is destroyed by viruses to form pores [10]. The nucleocapsid protein of the virus enters the columnar epithelial cells of the host midgut through envelope-mediated membrane fusion, triggering primary infection. The nucleocapsid protein enters the nucleus under the traction of actin, undergoes transcription, and completes the assembly of the progeny viral nucleocapsid in the nucleus [11]. The mature progeny nucleocapsid enters the cytoplasm through the nuclear pore, obtains the host cell membrane structure under the traction of capsid protein, completes the growth process, and forms a new progeny virus. Late during infection, the progeny ODV is re-embedded into the polyhedron and released into the environment after the death and disintegration of the host. In recent decades, extensive research has been conducted to enrich our understanding of the molecular mechanisms of silkworm resistance to BmNPV infection [12,13]. However, the molecular mechanism of its antiviral activity has not been fully elucidated.

Trehalose, also known as fungose, is a non-reducing disaccharide formed by connecting two glucose molecules via an α, α-1, 1-glucoside bond. It is found in a wide variety of organisms, including bacteria, fungi, insects, plants, and invertebrates [14,15]. Due to its unique chemical properties, trehalose has the advantage of protecting organisms from a variety of environmental stresses such as cold, oxidation, hypoxia, and drying [16]. Trehalose is the main hemolymph sugar of most insects [17], accounting for 80%-90% of the total hemolymph sugar content. It is synthesized in the adipose body, an organ similar to the mammalian liver and adipose tissue, and is released into the hemolymph [18,19]. Trehalose plays an important role in the growth and stress resistance of organisms, so some people call trehalose “the sugar of life”.

The trehalose metabolic pathways (synthesis, transport, and decomposition) have been extensively studied in insects. After insects feed, sucrose can be hydrolyzed into fructose and glucose in the gut [20]. Insects tend to ingest excessive amounts of sucrose, most of which is converted into long-chain oligosaccharides and excreted as honeydew [21]. The remaining sucrose is used for energy metabolism and the maintenance of osmotic balance [22]. Glucose is transported into the fat body via glucose transporters (GLUT) and participates in the synthesis of trehalose by trehalose-6-phosphate synthase/trehalose-6-phosphate phosphatase (TPS/TPP) [23]. Trehalose cannot directly cross the cell membrane [24], but depends on the specific trehalose transporter *TRET*1 for facilitated diffusion into the cell [25]. Kanamori et al., showed that the *tret*1 gene family is relatively conserved in insects, encoding proteins with different dynamic properties and participating in the release of trehalose from adipose bodies and its introduction into other tissues [26]. The *Tret*1 gene has been cloned from *Polypedilum vanderplanki*, *Anopheles gambiae*, and *Nilaparvata lugens*. Kikawada et al., isolated and characterized *Tret*1 from insects and found that the trehalose synthesized in the fat bodies was transported into the hemolymph [27]. Trehalose is hydrolyzed into two glucose monomers by alginase in the hemolymph and transported into tissues in the hemolymph to meet energy requirements [28]. Studies on trehalose transporters have mainly focused on energy metabolism and stress resistance, but there are few studies on the antiviral mechanisms underpinning insect trehalose transport [29].

In this study, we conducted transcriptomic profiling and bioinformatics analysis of the silkworm trehalose transporter *Bmtret*1 gene family and found it a candidate key gene family for silkworm BmNPV resistance in BmNPV susceptible species (Baiyu, BY). This information prompted us to analyze the expression of the sugar transporter gene in susceptible cultivars and its relationship to viral susceptibility. We also analyzed the homologous genes of *Bmtret*1 and their phylogenetic relationships to investigate their function in *Bombyx mori*. Using bioinformatic approaches, this study explores the functions of the silkworm *Bmtret*1 family, and provides a data reference for studying the molecular mechanisms behind insect virus resistance.

## 2. Results

### 2.1. Genome-Wide Identification and Phylogenetic Analysis of the Bmtret1s in B. mori

Based on the silkworm genome information, 21 *Bmtret*1 homologs were identified. These *Bmtret*1 homologs encode proteins of 204 to 591 amino acids with molecular weights of 23.18 to 65.47 kDa and theoretical isoelectric points of 4.83 to 9.48. Based on WoLF PSORT prediction of the subcellular localization, most of them (18/21) likely localize to the plasma membrane (PM) (Table 1). According to the amino acid sequence, each sample is divided into three clades, and samples within the same clade are highly related (Figure 1).

### 2.2. Chromosomal Localization of Bmtret1s

The target genes were mainly distributed on chromosomes 5, 7, 13, 14, 17, 20, 26, 27, and 28. Chromosome 27 has the most target genes, followed by chromosomes 20 and 26 (Figure 2). No active tandem genes and gene replication pairs were found in the preliminary screening, while the results need to be further verified.

### 2.3. Sequence Analysis of the Bmtret1s

Conserved regions in the *Bmtret*1 proteins were identified using multiple sequence alignment analysis of the amino acid sequence. The alignment and conserved motif analysis showed that the *tret*1 gene family retained four conserved sites, indicated by red highlighting (Figure 3).

### 2.4. Gene Organization and Promoter Analysis of Bmtret1s

Cis-acting elements are present in the peripheral sequences of genes that affect gene expression. Cis-acting elements include promoters, enhancers, regulatory sequences, and inducible elements, which are involved in the regulation of gene expression. The cis-acting element itself does not encode any protein, but merely provides an action site that interacts with trans-acting factors. According to the annotation information of the silkworm genome, the molecular characteristics of the *Bmtret*1 genes were analyzed, and a phylogenetic tree was constructed to identify possible functional elements in each gene. The *Tret*1’s exon was located, and the sample sequence was basically located in the exon region, suggesting that it was involved in the regulation of this gene expression. As can be seen in the distribution map of cis-elements in the *Tret*1 promoter region (Figure 4), stress-response-related elements were identified in multiple samples.

### 2.5. Expression Profile of Bmtret1s in Different Tissues of Silkworm

To determine the tissue and organ expression profile of *Bmtret*1s, the relative expression levels of the *Bmtret*1 genes in the silkworm hemolymph (HE), midgut (MG), fat body (FB), posterior silk gland (PS), and head (HD) were measured using qRT-PCR with silkworm actin 3 as the internal reference. The expression of BMSK0011446 in the phylogenetic branch in these tissues was relatively low. The gene expression patterns in phylogenetic branch I varied greatly, with BMSK0011410 having the highest expression in the midgut (MG); BMSK0011573 having the highest expression in the posterior silk gland (PS), hemolymph (HE), and posterior midgut (MG); and BMSK0011404 having the highest expression in the head (HD) (Figure 5a–c).

While the expression patterns of *Bmtret*1 in the phylogenetic clade is more diverse, the highest expressed *Bmtret*1s in the midgut (MG) were BMSK0003818, BMSK0015729, BMSK0015638, and BMSK0015673, where BMSK0015729 and BMSK0015638 had similar expression patterns, The expression levels of each tissue were ordered from high to low: midgut (MG), posterior silk gland (PS), head (HD), fat body (FB), and hemolymph (HE) (Figure 5d,q,s,t); BMSK0015120, BMSK0002683, and BMSK0002685 were the highest in the fat body (FB), with the latter two having similar expression patterns (Figure 5i–k). Four genes (BMSK0015774, BMSK0015674, BMSK0015633, and BMSK0015627) had the highest expression levels in the posterior silk glands (PS). With the exception of the high expression of BMSK0015627 in the midgut (MG), the expression patterns were very similar. All other tissues had relatively low expression levels (Figure 5g,n,p,r). The remaining genes have the highest expression in the head (HD). There are two patterns of expression. The expression pattern of BMSK0009966, BMSK0015122, and BMSK0007748 was similar to others in the phylogenetic branch I, except for having higher expression in the head (HD) and fat body (FB) (Figure 5e,f,o,c). The other type of *Bmtret*1s (BMSK0015118, BMSK0008304, and BMSK0012519) were the highest expressed in the fat body (FB) and midgut (MG) (Figure 5h,l,m).

### 2.6. Transcriptional-Level Responses of Bmtret1s to BmNPV Stress

The relative expression of *Bmtret*1 genes in the hemolymph (HE), midgut (MG), and fat body (FB) 24 h after infection with BmNPV was determined using qRT-PCR. BmNPV is a viral disease caused by infection with the polyhydrosis virus. All genes responded to infection with BmNPV. In the fat body (FB), the expression level of 16/20 *Bmtret*1s was increased and 4/20 was decreased. In the midgut (MG), 15/20 *Bmtret*1 genes were upregulated, 3/20 downregulated, and 2/20 unchanged. A different pattern was observed in the hemolymph, where 12/20 *Bmtret*1s were downregulated. Only one gene (BMSK0015729) showed an upregulated level in the hemolymph after infection (Figure 6).

Further, 9 out of the 20 genes (BMSK0011410, BMSK0011404, BMSK0003818, BMSK0015122, BMSK0015774, BMSK0015120, BMSK0002683, BMSK0002685, and BMSK0008034) had a similar expression pattern, being upregulated in both the midgut (MG) and the fat body (FB). These genes are also closely related in the phylogenetic tree (Figure 6a,c,d,f,g,i–l). With differences in their expression levels in the hemolymph (HE), the nine genes can be divided into two expression pattern groups. Five genes (BMSK0011404, BMSK0003818, BMSK0015122, BMSK0015120, and BMSK0002685) showed an insignificant response in the hemolymph (HE), while the remaining four genes were downregulated. Five other genes (BMSK0009966, BMSK0015118, BMSK0012519, BMSK0015627, and BMSK0015638) showed similar expression profiles to each other after BmNPV biological stress, with their expression levels being upregulated in the midgut and downregulated in the fat body (Figure 6e,h,m,r,s).

## 3. Discussion

Nuclear polyhedrosis virus disease of silkworms is highly infectious and harmful. It is the most common and most harmful silkworm disease in silkworm rearing and production, and causes serious economic losses every year [30,31]. Over the years, many researchers have been committed to screening and breeding resistant silkworm varieties and discovering resistance genes to elucidate the molecular mechanisms of silkworm resistance to BmNPV [32]. With the development of biotechnology, more achievements have been made in the study of *B. mori*’s resistance to the BmNPV virus at the molecular level, and further studies have been made on genes or proteins that may be involved in the antiviral mechanism. After the completion of silkworm genome sequencing, gene chip technology has become an important gene expression analysis method, which has the advantages of a large throughput and high accuracy. Zhou et al., detected 92 differentially expressed genes in the intestinal tissues of silkworm varieties BC8 and 306 after 12 h of toxic treatment with nucleic acid probes. They further analyzed 10 upregulated genes using fluorescent quantitative PCR. Fluorescence quantitative PCR technology can quickly compare and analyze the expression of all genes in the sample [33]. *BmS3A* is related to the inhibition of the apoptosis of infected cells, which inhibits the replication of viruses [34]. The *SOP*2 gene may promote the actin polymerization process and affect virus replication [35]. We selected the midgut tissues of a conventional susceptible strain of silkworm Baiyu infected with BmNPV for second-generation RNA-Seq transcriptome sequencing, for systematic screening of candidate differentially expressed genes involved in BmNPV infection resistance in the *Bmtret*1 gene family.

Trehalose, as a type of natural sugar, can be used as a protective factor to protect the organism from external environmental stresses or internal metabolic disorders. TRET, the trehalose transporter, can transport trehalose from the fat body to the hemolymph, and plays an important role in insect stress resistance [26,27]. While TRET plays an important role in the resistance to numerous insect stresses, there are very few studies on the effects of *TRET* on virus infection. Some studies have speculated that the trehalose transprotein-1 (*Tret*1) gene may be related to the transport of the virus during the interaction between gray planthoppers and the rice stripe virus [36]. In addition, the trehalose transprotein-1 (*BmTret*1-like) gene of silkworm plays a specific role in the mechanism of BmNPV virus resistance [37]. Recent studies have shown that the expression of the *BmTret*1-X1 gene has a clear inhibitory effect on the expression of viral genes in BmNPV [38]. The transcriptome results found that the expression level of *Bmtret*1s significantly responded to BmNPV biological stress, and we speculated that *Bmtret*1s may play an important role in the infection of BmNPV.

In this study, it was found that the *Bmtret*1 gene family varied greatly in different tissues with possible functional differences. Because trehalose is involved in the process of silkworm epidermis formation, it is speculated that the high expression level in the head may be associated with ecdysone and juvenile hormone. The higher expression in the two detoxifying organs of the midgut and fat body indicates that *Bmtret*1s may participate in the molecular mechanisms of disease resistance. In BmNPV-susceptible varieties of white jade silkworm, the vast majority of *Bmtret*1 genes are downregulated in response to BmNPV oral infection. We speculate that the downregulation of the trehalose transporter gene in the hemolymph allows for BmNPV invasion and is the cause of susceptibility. Furthermore, the high expression of *Bmtret*1s in the midgut and fat body correlates with viral resistance in these two detoxification organs. Some of the genes being downregulated in the detoxification organs suggest that not every gene in the *Bmtret*1 gene family is involved in the defense against BmNPV.

## 4. Materials and Methods

### 4.1. Sericulture Breeding and Virus Preparation

The Baiyu were acquired from the Resources Center of the Silviculture Research Institute, the Chinese Academy of Agricultural Sciences. Baiyu silkworms have a strong desire to eat mulberries, have a larger body and a bluish–white color, and are susceptible to BmNPV. All silkworm larvae were raised on fresh mulberry leaves. The worms were raised at 27 ± 1 °C at 75 ± 5% relative humidity under a 12 h day and night cycle. The BmNPV strain was maintained in our laboratory and purified according to the protocol reported by Rahman et al. [8]. After starvation treatment, the experimental group was fed 7 μL of BmNPV suspension (2 × 10^8^ polyhedra/mL), and the control group was fed normally. The hemolymph, midgut, and fat body of the larvae from both groups were taken 24 h after infection. After 72 h, the hemolymph, midgut, fat body, posterior silk gland, and head of the control group were collected. There were three biological replicates, and five silkworms constitute one biological replicate. They were taken and stored at −80 °C after infection.

### 4.2. Identification of the Bmtret1 Gene Family in B. mori

Sequences homologous to the *Bmtret*1 genes were downloaded from the silkworm genome database, SilkDB 3.0 (https://silkdb.bioinfotoolkits.net/, accessed on 1 May 2023). Their chromosomal distribution and homology relationships were analyzed. The Biological Toolbox v1.098774 was used to analyze the sequence length, molecular weight, and theoretical isoelectric point (PI) values for each homologous gene. The distribution of TM helices was determined using TMHMM Server v.2.0 (http://www.cbs.dtu.dk/services/TMHMM/, accessed on 18 May 2023) Forecast. Subcellular localization of the *Bmtret*1s protein was predicted using the online tool WoLF PSORT (https://wolfpsort.hgc.jp/, accessed on 21 May 2023).

### 4.3. Chromosomal Localization and Homology Analysis of Bmtret1s

The chromosomal location information of the *Bmtret*1 gene family was extracted from the silkworm genome annotation file. This operation was performed and visualized in Tbtools v1.098774.

### 4.4. Sequence Alignment

The *Bmtret*1 protein sequences were aligned using clustalW and assembled in MEGA 11.0.

### 4.5. Gene Structure and Promoter Analysis of Bmtret1s

The gene structure of each *Bmtret*1 was displayed based on the genome sequence and its annotation file using Gene Structure View assembled in Tbtools v1.098774. The upstream 2000 bp sequences were extracted for the promoter region analysis. The cis-acting elements were predicted using PlantCARE (http://bioinformatics.psb.ugent.be/webtools/plantcare/html/, accessed on 25 May 2023).

### 4.6. Phylogenetic Analysis of Bmtret1s

A neighbor-joining (NJ) phylogenetic tree was constructed using full-length *Bmtret*1 protein sequences from *B. mori* and using MEGA 11.0 and JTT + G models, with bootstrap tests with 1000 replicates.

### 4.7. RNA Extraction and Quantitative Real-Time PCR (qRT-PCR) Analysis

According to the manufacturer’s instructions, the EASYspin Tissue/Cell RNA Rapid Extraction Kit (Aidlab, Beijing, China) and HiScript III 1st Strand cDNA Synthesis Kit (+gDNA wiper) were used for the RNA extraction and cDNA synthesis.

The ChamQ SYBR qPCR Master Mix from Vazyme was used. *Actin*3 was used as a reference gene. The RT-qPCR was performed using an ABI StepOnePlus™ Real-Time PCR System (Shanghai, China) to verify the expression patterns of *Bmtret*1 in different tissues at different times. GraphPad Prism 8.0 was used to visualize the RT-qPCR results and to perform Duncan’s test and Student’s *t*-test. In Figure 5, a *p*-value <0.05 was marked as significant. In Figure 6, a *p*-value < 0.001 was marked with ***, *p*-value < 0.01 was marked with ** and *p*-value < 0.05 was marked with *. At least three biological replicates were performed, respectively, for the RT-qPCR. The primers used are shown in Appendix A Table A1. The relative expression level of these genes was estimated according to the 2^−ΔΔCt^ method.

## 5. Conclusions

We conducted transcriptome and phylogenetic analysis of the silkworm *Bmtret* gene family and performed expression profiling and transcript-level analysis after infection with BmNPV. The *Bmtret*1 gene family has been implicated in silkworm resistance against BmNPV, and the high expression of most *Bmtret*1s in the midgut and fat body may inhibit the gene transcription of BmNPV and DNA replication, and thus reduce the assembly efficiency of virions to resist BmNPV infection. The *Bmtret*1 gene family of silkworm trehalose transporter was preliminarily identified as a key candidate gene family of silkworm BmNPV resistance, and the specific mechanism needs further study.

## Figures and Tables

**Figure 1 ijms-25-00402-f001:**
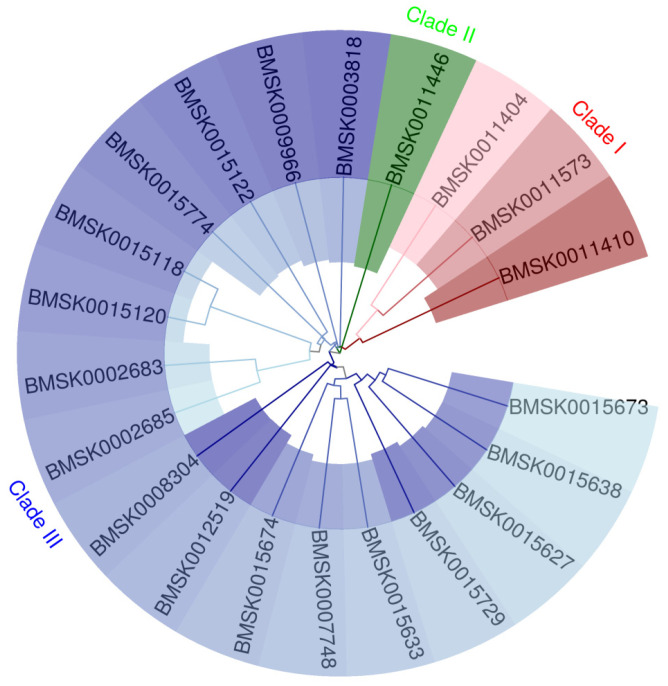
Phylogenetic relationships of the *TRET*1 family genes in *Bombyx mori*. The sequences of the 21 *TRET*1 proteins from the above insect were aligned using Clustal Omega, and the phylogenetic tree was constructed using MEGA 11.0 using the NJ method with 1000 bootstrap replicates.

**Figure 2 ijms-25-00402-f002:**
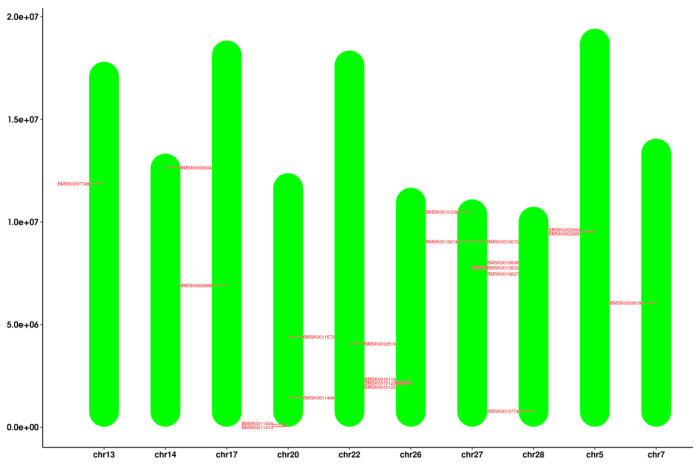
Distribution of *Bmtret*1 genes in *Bombyx mori* chromosomes. The scale is provided in megabases (Mb).

**Figure 3 ijms-25-00402-f003:**
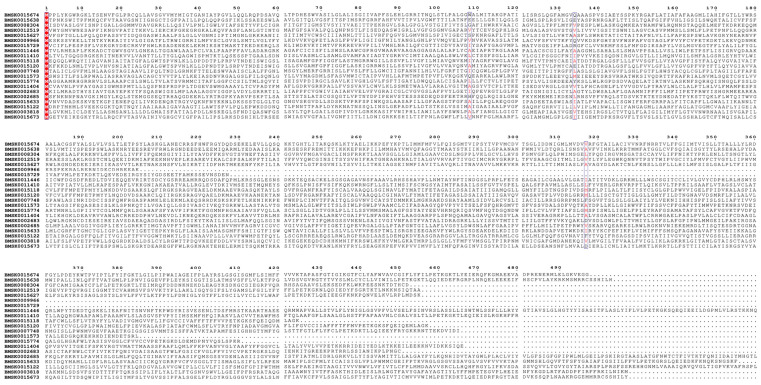
Multiple sequence alignment of *Bmtret*1 proteins. Conserved sites are indicated by red highlights.

**Figure 4 ijms-25-00402-f004:**
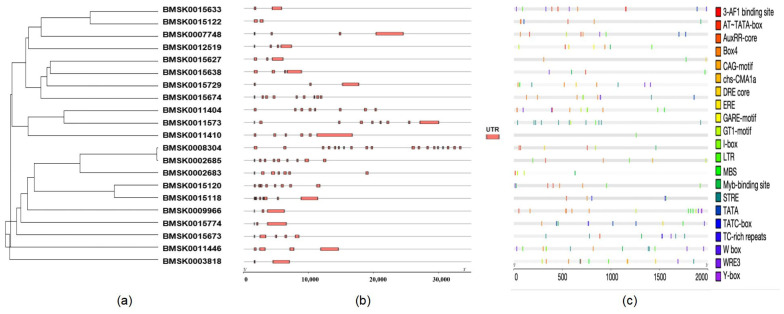
Gene organization of *Bmtret*1s and cis-elements in promoter regions of *Bmtret*1s. (**a**) Phylogenetic tree using 21 *Bmtret*1s. (**b**) Exon/intron structures of *Bmtret*1s. (**c**) Cis-element distribution in the promoter regions of *Bmtret*1s.

**Figure 5 ijms-25-00402-f005:**
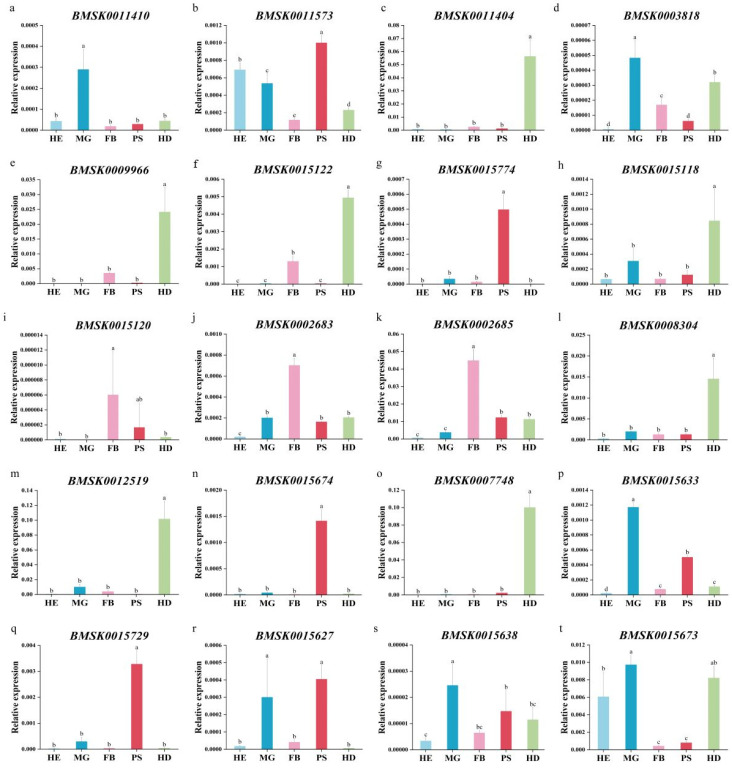
Transcript levels of *Bmtret*1s in hemolymph, midgut, fat body, posterior silk gland, and head of *Bombyx mori*. Three biological replicates were analyzed. Different letters indicate statistically significant differences (Duncan’s test, *p* < 0.05). Values are presented as mean ± SEM. The first three genes belong to branch I, and the expression level of gene of branch II is relatively low. The other genes belong to branch III. (**a**) BMSK0011410; (**b**) BMSK0011573; (**c**) BMSK0011404; (**d**) BMSK0003818; (**e**) BMSK0009966; (**f**) BMSK0015122; (**g**) BMSK0015774; (**h**) BMSK0015118; (**i**) BMSK0015120; (**j**) BMSK0002683; (**k**) BMSK0002685; (**l**) BMSK0008304; (**m**) BMSK0012519; (**n**) BMSK0015674; (**o**) BMSK0007748; (**p**) BMSK0015633; (**q**) BMSK0015729; (**r**) BMSK0015627; (**s**) BMSK0015638; (**t**) BMSK0015673.

**Figure 6 ijms-25-00402-f006:**
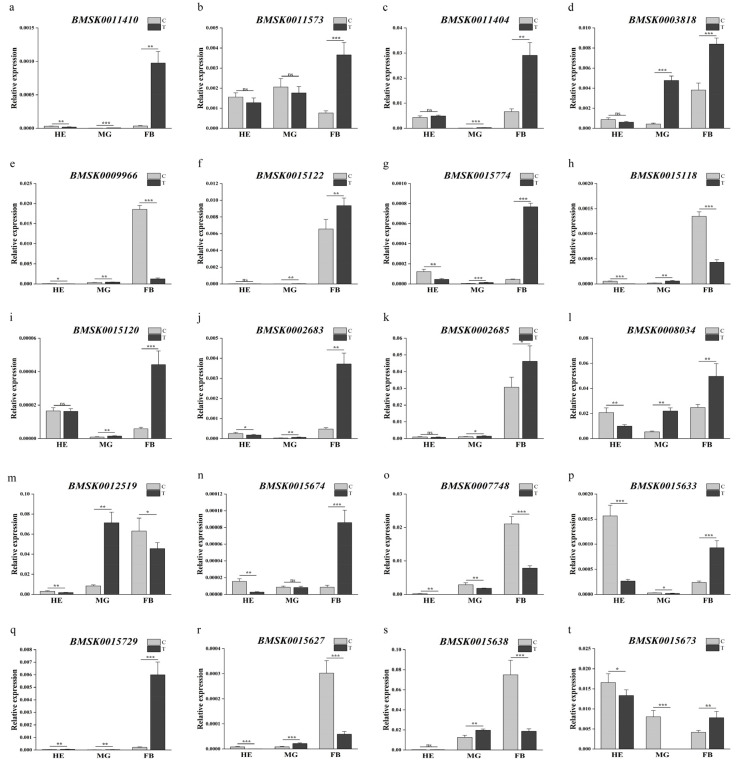
Expression levels of 20 *Bmtret*1 genes in response to BmNPV stress conditions. Three biological replicates were analyzed. Asterisks indicate significant differences as determined using Student’s *t*-test (* *p* < 0.05; ** *p* < 0.01; *** *p* < 0.001). Values are presented as mean ± SEM. C is the control group and T is the treatment group. The first three genes belong to branch I, and the expression level of gene of branch II is relatively low. The other genes belong to branch III. (**a**) BMSK0011410; (**b**) BMSK0011573; (**c**) BMSK0011404; (**d**) BMSK0003818; (**e**) BMSK0009966; (**f**) BMSK0015122; (**g**) BMSK0015774; (**h**) BMSK0015118; (**i**) BMSK0015120; (**j**) BMSK0002683; (**k**) BMSK0002685; (**l**) BMSK0008304; (**m**) BMSK0012519; (**n**) BMSK0015674; (**o**) BMSK0007748; (**p**) BMSK0015633; (**q**) BMSK0015729; (**r**) BMSK0015627; (**s**) BMSK0015638; (**t**) BMSK0015673.

**Table 1 ijms-25-00402-t001:** *TRET*1 gene family in *Bombyx mori*.

Gene ID	CDS Size (bp)	Protein Physicochemical Characteristics				TMHs	Subcellular Localization *
		Length (aa)	MW (kDa)	pI	Aliphatic Index		
BMSK0011410	1443	480	51.74	9.31	116.67	12	PM
BMSK0011573	1155	384	42.90	6.02	107.16	7	PM
BMSK0011404	1401	466	50.77	8.31	111.33	12	PM
BMSK0011446	1632	543	58.85	7.84	100.72	9	PM
BMSK0003818	1524	507	56.44	7.55	112.76	11	PM
BMSK0009966	615	204	23.18	8.28	101.18	4	EX
BMSK0015122	1635	544	58.66	9.48	113.86	11	PM
BMSK0015774	1233	410	44.81	8.17	108.24	10	PM
BMSK0015118	1275	424	46.07	4.83	115.26	11	PM
BMSK0015120	1374	457	49.34	6.59	117.13	12	PM
BMSK0002683	1353	450	49.19	8.61	116.42	11	PM
BMSK0002685	1776	591	65.47	9.15	98.65	10	PM
BMSK0008304	1359	452	49.90	8.74	102.23	10	PM
BMSK0012519	1368	455	49.99	9.42	100.26	10	PM
BMSK0015674	1494	497	54.72	9.05	107.95	10	MT
BMSK0007748	1398	465	51.74	9.05	108.04	12	PM
BMSK0015633	1608	535	58.76	8.92	103.20	12	PM
BMSK0015729	684	227	26.00	5.21	92.69	2	CY
BMSK0015627	1368	455	50.32	9.08	125.34	11	PM
BMSK0015638	1512	503	56.54	9.28	104.10	12	PM
BMSK0015673	1515	504	55.87	9.08	113.57	10	PM

* The subcellular localizations were predicted using WoLF PSORT. MT, Mitochondrial; PM, Plasma Membrane; EX, Extracellular; CY, Cytoplasmic.

## Data Availability

The data presented in this study are available in Appendix A.

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
