# Peer review of "The Bmtret1 Gene Family and Its Potential Role in Response to BmNPV Stress in Bombyx mori"

_ijms, 2023, doi:10.3390/ijms25010402_

Round 1

Reviewer 1 Report

Comments and Suggestions for Authors

The manuscript by Lin et al provides a comprehensive overview of the Bmtret1 gene family and its potential role in stress resistance in Bombyx mori. The data is convincing and supports the conclusions. The study is also well-explained.

There are some minor issues the authors need to address:

1. The authors need to provide an better explanation of the connection of Bmtret1 gene family and BmNPV infection.

2. The authors need to provide more details about the qPCR experiments such as the reagents used and the number of replicates. Also, they need to have a separate section in the methods section to explain how statistical tests were performed.

Reviewer 2 Report

Comments and Suggestions for Authors

Lin eat al conducted transcriptomic profiling and bioinformatics analysis of the silkworm trehalose transporter Bmtret1 gene family and found a candidate key gene family for silkworm BmNPV resistance in BmNPV susceptible species. The manuscript reads well. However I have a couple of concerns.

Figure 3, how did the authors quantify the conserved regions. Also what are the significance of these results?

Figure 4, Not clear to me how the authors identified cis-acting elements.

Figure 5 and 6, you need biological replicates rather than technical replicates. No matter how many technical replicates you get, it will never be convincing.

Figure 6 What are C and T?

I think insect blood should be called haemolymph.

Reviewer 3 Report

Comments and Suggestions for Authors

Overall Summary: In this manuscript, Lin et al. examine the occurrence, variation, and transcriptional response of trehalose related Bmtret1 genes in Bombyx mori in response to BmNPV interaction. While the authors do explain why the focus of the study was on this one gene, the expression patterns don’t clearly implicate the gene consistently during viral interactions. Also, there are issues with some phylogenetic comparisons done in the study and there is a lack of any other gene being used to compare in terms of transcriptional response during infection, considering the complexity of this process. Specific comments are listed below.

Specific Comments:

·         The abstract needs to include some background of Bmtret1, etc. The organization of the abstract also needs to have some introduction followed by rationale and then results.

·         Check the use of prepositions in the abstract.

·         Results 2.1: The process by which the clades were determined needs to be stated here along with the outgroup that was used.

·         Figure 3: the legend of the figure needs to state what exactly is highlighted here. Also, for figure 3, the explanation in 2.3 states that the conserved motifs are highlighted, however the highlighted residues seem to show variation. Also, it states that four regions are highlighted, but one of them is just the start codon.

·         Figure 4: this figure needs to be organized and labelled. A, B and C panels have tiny font that can’t be read. This needs to be increased. Also, the gene and the surrounding elements need to be differentiated by color coding. 4C has a lot of color-coded elements on the side, but the main panel showing the distribution of the elements is not marked, so it is unclear where the elements in different colors are located.

·         Figure 5 needs to have a larger font, since most of it is not clearly visible. The different phylogenetic branches need to be highlighted or marked in figure 5 to show the grouping. Also, if the relative expression was analyzed, what is the control conditions that all the other conditions were compared with? This needs to be clarified in the results.

·         The font in figure 6 also needs to be larger. Also, in the case of figure 6, what is the normalization condition that was used to determine the relative expression. This needs to be explained.

·         Figure 6 doesn’t clearly indicate that Bmtret1 genes are always correlated with BmNPV stress. The results are mixed a lot of times with variable expression in different tissue and the control condition having greater expression many times. This needs to be stated in the results and discussion.

·         The discussion section doesn’t explain much of the data collected in the manuscript and only provides a review of other studies similar to the introduction. The variability in results needs to be explained in detail.

·         Further, the discussion section mentioned that a previous transcriptomic study by Zhou et al. found several differentially expressed genes in silkworm viral interactions. However, it is unclear why this study did not include any other gene as an alternative to the Bmtret1 gene when examining the transcriptional changes during viral interactions. As a control, other genes shown to have an impact during infection should be examined via qPCR or, a transcriptomic study should be included by a subset of samples and tissue types. This will show the impact of different genes during the insect-viral interactions. This is required since there is no clear evidence that Bmtret1 is the only affected target.

·         In the methods sections, the background of the various silkworm larva types used is not provided.

·         All open-source bioinformatics tools should have citations.

·         qPCR data analysis and normalization are not described in terms of the control condition and the analysis method.

Comments on the Quality of English Language

·         Check the use of prepositions in the abstract.

Round 2

Reviewer 2 Report

Comments and Suggestions for Authors

Figure 3, how did the authors quantify the conserved regions. Also what are the significance o f these results? Answer: The conserved region set is the number of repeated amino acids exceeding one. We wanted to explore whether there are conserved functional areas in the Bmtret1 protein family, but unfortunately not.

OK.

Figure 4, Not clear to me how the authors identified cis-acting elements. Answer: We will add the following in the materials and methods: Gene structure and promoter analysis of Bmtret1s The gene structure of each Bmtret1 was displayed based on the genome sequence and its annotation file using Gene Structure View assembled in Tbtools v1.098774. The upstream 2000 bp sequences were extracted for in promoter region analysis. Cis-acting elements were predicted using PlantCARE (http://bioinformatics.psb.ugent.be/webtools/plantcare/html/).

OK.

Figure 5 and 6, you need biological replicates rather than technical replicates. No matter how many technical replicates you get, it will never be convincing. Answer: Extracted RNA was derived from tissue mixed samples from five organisms of the same group, which can eliminate individual differences.

No you can't. You know nothing about the variance of biological difference by your methods. You need to create three different RNA extract and perform the independent experiments. Otherwise you cannot deny that you got a single lucky data point from a single extract.

Figure 6 What are C and T? Answer: C is the control group and T is the treatment group, that is, the silkworm group with BmNPV.

OK, you need to reflect it in your manuscript.

I think insect blood should be called haemolymph. Answer: All right, it has been modified following your suggestion.

OK.

Reviewer 3 Report

Comments and Suggestions for Authors

The authors have addressed the comments. 

Figure 4 might need slightly larger font on the legends. 

Round 3

Reviewer 2 Report

Comments and Suggestions for Authors

As I said, you need to take three biological replicates but not technical replicates to support your conclusion. Without the data I have to reject your manuscript.

Author Response

Thank you very much. In fact, we have done three biological replicates in our previous experiment, and we have revised the description of experimental methods in the latest revised manuscript (see the attachment).